# Constructing a Reference Genome in a Single Lab: The Possibility to Use Oxford Nanopore Technology

**DOI:** 10.3390/plants8080270

**Published:** 2019-08-06

**Authors:** Yun Gyeong Lee, Sang Chul Choi, Yuna Kang, Kyeong Min Kim, Chon-Sik Kang, Changsoo Kim

**Affiliations:** 1Department of Crop Science, College of Agricultural and Life Sciences, Chungnam National University, Daejeon 34134, Korea; 2National Institute of Crop Science, Rural Development Administration (RDA), Wanju 55365, Korea

**Keywords:** Keywords: sorghum, Canu, Miniasm, MinION, long-read sequencing

## Abstract

The whole genome sequencing (WGS) has become a crucial tool in understanding genome structure and genetic variation. The MinION sequencing of Oxford Nanopore Technologies (ONT) is an excellent approach for performing WGS and it has advantages in comparison with other Next-Generation Sequencing (NGS): It is relatively inexpensive, portable, has simple library preparation, can be monitored in real-time, and has no theoretical limits on reading length. *Sorghum bicolor* (L.) Moench is diploid (2*n* = 2*x* = 20) with a genome size of about 730 Mb, and its genome sequence information is released in the Phytozome database. Therefore, sorghum can be used as a good reference. However, plant species have complex and large genomes when compared to animals or microorganisms. As a result, complete genome sequencing is difficult for plant species. MinION sequencing that produces long-reads can be an excellent tool for overcoming the weak assembly of short-reads generated from NGS by minimizing the generation of gaps or covering the repetitive sequence that appears on the plant genome. Here, we conducted the genome sequencing for *S. bicolor* cv. BTx623 while using the MinION platform and obtained 895,678 reads and 17.9 gigabytes (Gb) (ca. 25× coverage of reference) from long-read sequence data. A total of 6124 contigs (covering 45.9%) were generated from Canu, and a total of 2661 contigs (covering 50%) were generated from Minimap and Miniasm with a Racon through a *de novo* assembly using two different tools and mapped assembled contigs against the sorghum reference genome. Our results provide an optimal series of long-read sequencing analysis for plant species while using the MinION platform and a clue to determine the total sequencing scale for optimal coverage that is based on various genome sizes.

## 1. Introduction

The whole genome sequencing (WGS) has become a crucial tool for understanding genome structure and genetic variation. Next-generation sequencing (NGS) technology, which has been actively used over the past decade, has revolutionized the genomic research of plants as well as animals and microorganisms, which consequently leads to a high-throughput WGS [1,2]. However, most of the existing NGS techniques typically generate short-reads (35–700 bp) and the assembled sequences from these short-reads have resulted in an occurrence of gaps. This is because short-reads are not able to span repetitive sequences longer than their length due to the limitations of assembly completeness, which thereby causes an incomplete genome assembly [1,3].

Unlike NGS, the third-generation sequencing (TGS) technology enables the generation of long-reads as a single molecule by preserving the native DNA state as much as possible during library construction and performing sequence detection through electrical or optical signals [2,3,4]. The major advantage of long-read sequencing is that it may be able to resolve the gaps that occurred from short-read assemblies [5]. Although the TGS market is overwhelmingly controlled by Pacific Biosciences (PacBio), the MinION platform [4] of Oxford Nanopore Technologies (ONT) is relatively inexpensive in comparison with other NGS platforms and allows the production of long-reads by only using small portable devices. In addition, the simple preparation of a sequencing library does not require specific large instrumentation and complicated library preparation [2,6]. The MinION platform [4] is able to monitor the progress of the sequencing reaction in real-time as well as directly detect nucleotide modifications. As a result, this platform may be desirable for a small-scale laboratory to run and manage it in-house. Moreover, there is no theoretical limit on the read length, so they could obtain a read sequence that has several hundred kilo base pairs (Kbp) or more if a high-molecular weight (HMW) genomic DNA (gDNA) and sequencing library were properly prepared [4,7]. If the high-error rate can be overcome, Nanopore sequencing may be very useful for the *de novo* assembly or for studying the structural or single-nucleotide variations [4].

*Sorghum bicolor* (L.) Moench is one of the most consumed crops in the world and it represents the C4 model plant. The WGS for sorghum has been performed and publicly is available [8]. Sorghum is diploid (2*n* = 2*x* = 20) with a genome size of about 730 Mb and a repeat content of ~61% (from homozygous sorghum genotype BTx623) [8,9]. Currently, the genome sequence information can be found in the Phytozome database (https://www.phytozome.net/, [10]) and it is being continuously updated. However, to date, even though 4426 gaps were closed, and the overall contiguity increased by 5.8× in a recent update (*S. bicolor* v3.1.1 in Phytozome database), consequently sorghum still remains an incomplete genome sequence. It is difficult to complete genome sequencing for plant species, since plant species have a more complex genome structure and larger genome size than animal species [11]. Plants have evolved through expanding or altering genomes, for example, the whole genome duplication, as a way of adapting to the external environment due to sessility, which results in a lot of repeated sequences [11,12,13]. During the evolutionary process, factors, such as polyploidy, repetitive sequences, heterozygosity, and transposable elements, have contributed to the plant genome size and complexity [11,14].

Recently, Nanopore sequencing while using the MinION platform has been applied in various fields for plant species, but it remains somewhat limited. The detection of transposable elements associated structural variants (TEASVs) in *Arabidopsis* [15], the validation of assemblers for the *Arabidopsis* genome [16], the *de novo* assembly of the *Solanum pennellii* genome through hybrid sequencing [17], the identification of novel genes that are related to nucleotide-binding leucine-rich repeat (NLR) [18], the improvement of maize reference genomes [19], and the field-based analysis for identifying closely-related plants (*Arabidopsis* spp.) [20] are some examples of this application. Moreover, apart from recent improvements in the accuracy of the Nanopore sequencing, there is a trend in improved the accuracy of assembled sequences by bioinformatically compensating long-reads using short-reads, which leads to the obtaining of a high-contiguity genome assembly [21,22].

In this study, we conducted the genome sequencing for *S. bicolor* cv. BTx623 while using the MinION platform [4] and obtained 895,678 in the read number and 17.9 Gbp (ca. 25× coverage of the entire sorghum genome) from the long-read sequence data. We performed *de novo* assembly while using two different tools and mapped the assembled contigs against the sorghum reference genome to determine how much the MinION sequencing results cover the entire genome. As a result, from Canu [23], a total of 6,124 contigs (344,453,188 bp in length covering 45.9% of reference) were generated, and from Minimap and Miniasm [24] with five rounds of Racon [25] polishing tool, a total of 2661 contigs (375,105,174 bp in length covering 50% of reference) were generated. Our results provide an optimal series of long-read sequencing analysis for plant species while using the MinION platform [4] and a clue for determining the total sequencing scale for optimal coverage based on various genome sizes in order to obtain satisfactory results for the *de novo* assembly.

## 2. Results

### 2.1. MinION Sequencing of Sorghum Accession BTx623 Genome

We conducted the sequencing of sorghum HMW gDNA by using the MinION platform [4] to assess the high quality *de novo* assembly for the sorghum genome (cv. BTx623). The summary statistics for each run were separately calculated and combined into one table (Table 1). We constructed three libraries: DNA fragmentation was performed (around 20 Kbp) in one of the three libraries (2nd in Table 1), and the remaining libraries used an intact HMW gDNA. MinION sequencing for each library was conducted while using the standard script that was provided in the MinKNOW software. The total yielded amount of sequencing data varied among 2.85 Gb, 11.71 Gb, and 3.34 Gb, with different initial HMW gDNAs for library preparation. The second result generated the largest data size when compared to the other two results (first and third). since it used fragmented HMW gDNA. A total of 17.9 Gb of raw reads were generated, representing 25× of the total sorghum genome (based on 730 Mb). Overall, the longest read length was up to 110 Kbp, while the most abundant reads were in the range of 908 bp to 1028 bp in length.

The raw sequences were aligned to the sorghum BTx623 reference genome while using the BWA-mem version 0.7.15 [26] with a default option. All of the raw reads were separately analyzed and combined before downstream processing. The average depth was approximately 8.6× and the mapping rate was 97% for the combined data (Table 2). The Q-score was around 10, which indicated that a read error rate should be around 10%. The coverage distribution was plotted while using the Mosdepth [27] output (Figure 1). The depth of coverage calculation results from both the SAMtools [28] and Mosdepth [27] showed that only 7.0× to 8.56× of the sorghum genome were covered by using the combined sequencing data (17.9 Gb in total) generated from the three libraries. However, the mapping rates were more than 97%, indicating that the sequencing data generated from the MinION platform could contain redundant coverage in the specific regions of the sorghum genome. This redundant coverage was particularly concentrated in regions presenting highly repeated DNA contents. Furthermore, in many cases, certain regions are difficult to sequence and/or map because of repetitive DNA or sequences that were aligned to multiple places in the genome. We will need additional data to resolve these problems.

### 2.2. Assembly Results Using Canu

The processed raw reads (from UniTigging/READs) were *de novo* assembled while using Canu (version 1.6) [23]. The correction step in the Canu assembly [23] improves the accuracy of each read base by building read and overlapped databases and choosing overlaps for correction. The corMaxEvidenceErate = 0.15 parameter is suggested from the Canu documentation for the AT/GC rich eukaryotic genome. Therefore, this parameter was added to run our data analysis and other options were used as a default.

A total of 9.4 Gb out of 17.9 Gb raw reads were loaded due to the specific feature that the low coverage data less than 10× in any region are eliminated by the Canu [23] program. Only 8.0 Gb (11.56×) remained after the correction step (Table 3). The correction phase improved the accuracy of bases in the reads, while the trimming phase cleaned the reads to the portion that appeared to be a high-quality sequence, as well as removed suspicious regions such as the remaining SMRTbell adapter. However, this was only applicable to the PacBio data. Therefore, the trimming phase may not drastically affect the entire read contents for MinION [29] trials. The final assembly phase ordered the reads into contigs and then generated consensus sequences (unitigging concensus sequence). The final unitigging concensus sequence length was about 344 Mbp (344,366,012 bp), with N50, 98 Kbp (97,987 bp).

### 2.3. Assembly Results Using Minimap, Miniasm, and Racon

The raw read overlapper, Minimap [24], was used to find overlaps, and Miniasm [24] was used to complete *de novo* assemblies while using the Minimap [24] results (Table 4). They directly produce unpolished and uncorrected contig sequences from the overlaps of raw reads. As a result, polishing steps should be indispensable to improving their credibility. The five rounds of Racon [30] were used to correct raw contigs to produce better quality sequences. The file size differences between the raw file (17.9 Gb) and Minimap (13.2 Gb) indicated that our combined data had about 4.7 Gb file size of duplicated overlaps. By using 13.2 Gb size of raw read overlaps, the unpolished and uncorrected contig sequences with a file size of 368 Mb and a contig length of 370 Mbp were generated. The final length of consensus sequences for the three combined data sets after five rounds of Racon [30] polishing steps was about 375 Mbp with a N50 value of 199 Kbp (Table 4). In this consensus sequence, the longest contig was 1 Mbp in length and the shortest contig was 779 bp. The final sequence length for the combined data from the Racon results (375,105,174 bp) (Table 4) was slightly longer than that of the Canu [23] result (344,366,012 bp) (Table 3).

### 2.4. Confirmation of de Novo Assembly

To visualize the alignments between the assembled sequences from Miniasm [24] and each of the sorghum reference chromosomes, the mummerplot option from the Mummer software version 3.0, [31] with default parameters (Figure 2) was used. A total of 2661 contigs from 375,105,174 bp in length were generated after undertaking polishing steps five times with Racon [30]. The *x*-axis represents each chromosome of reference and the *y*-axis represents 2261 contigs. A perfect alignment between the contigs and each chromosome would completely fill the positive diagonal (slope == 1), while a line of slope == −1 represents an inverted segment of conservation between the two sequences. In chromosome 2, the contigs were not aligned either forward or reverse in some parts of the chromosome. In other chromosomes, the contigs tended to partially align to chromosomes, either forward or reverse (Figure 2). It was almost impossible to confirm the alignment trend between the Canu [23] consensus sequences and sorghum reference, since the Canu [23] generated relatively short and almost three times more contigs (6124 contigs) than Racon (2661 contigs) [30], which used the mummerplot (refer to Table 5 for comparison). Nevertheless, there are some genomic regions that cannot be covered by assembled contigs with those two bioinformatics tools, which indicates that more than 25X coverage data are required for the *de novo* assembly of sorghum genome.

## 3. Discussion

### 3.1. Optimization of Genome Assembly by Using Different Assemblers

We performed MinION sequencing [29] for the sorghum and compared the final results that were obtained from two different *de novo* assemblies against the reference genome in this study. With the ca. 12.63× coverage raw reads, the *de novo* assemblies from Canu [23] and Miniasm [24] showed 0.459× and 0.5× coverages for the sorghum genome, respectively. In other words, the completion of the *de novo* assembly can be mathematically achieved by increasing the amount of raw data to more than 25× for the sorghum example. However, the quality of the *de novo* assembly can be affected by a plethora of factors, such as the contents of genes, GC ratio, and the length of repetitive sequences, genome size, and ploidy numbers. We once tried to formulate the relationship between the minimum coverage that is required for the *de novo* assembly and the amount of raw reads. However, it was not feasible due to various factors, such as genome size, the contents of repeated sequences, the preparation of HMW DNA samples, and random sequencing errors. Therefore, the amount of long-read data needed for the *de novo* assembly should be empirically determined by stacking up the size of data when it is required. For example, one can use PacBio technology to generate long-read information; however, it requires a large-scale experiment with relatively high costs. On the contrary, the MinIon platform can be performed cell by cell with reasonable costs. In consequence, our main idea is to show that the *de novo* assembly for any species that does not have reference sequences can be performed in a single laboratory with cost-effective ways, owing to the newly developed ONT apparatus. In addition, a comparison between the two representative long-reads assemblers, Canu [23] and Miniasm [24], indicates that Miniasm [24] with Racon [30] correcting steps provides better assembly in terms of the number of contigs and N50 values.

Depending on the type of assembler, different assembly results may be obtained for the genome of the plant species [16,32]. As aforementioned, we showed the *de novo* assembly results while using Canu and Miniasm (with Minimap and Racon) for the sorghum genome. However, it is imperative to determine which assembler is suitable for optimal results when considering the genome size, structural variation, and genome complexity of the genome in the genome assembly of a particular plant species. In addition, bioinformatic efforts should be used to ensure that the misassembled or ambiguous sequences, such as repetitive regions of the genome, gaps, discrimination between paralogues and alleles or between genes and pseudogenes [11], are properly assembled. RNA sequencing using the MinION platform can be used to identify the isoforms of the transcripts. Through this, various isoforms have been identified without any assembly process, followed by non-redundant isoform clustering. The resulting information can be used for genome annotation and, consequently, can be integrated to increase the contiguity and accuracy of the results from the existing genome assembly [5].

### 3.2. Advantages of Current Combinational Sequencing

There is still not a lot of research in plant species because of its genome complexity as compared with studies on microorganisms or animal species. Genome sequencing has continued to develop through the classical Sanger method and NGS to TGS. Plant species have been actively studied in this process, but already-sequenced plant species have genomes with low complexity and that are of relatively small size until the advent of NGS [11]. NGS technology makes it possible to perform sequencing, regardless of genome size, which is a substantial technical breakthrough in overcoming these limitations. However, sequencing for plant species with large and complex genomes was not resolved in terms of contiguity and accuracy due to the limitations of the short-read assembly. In this respect, TGS that produces long-reads can be an excellent tool. For instance, wheat (*Triticum aestivum*) has a genome size of about 15 Gb, an allohexaploid (2*n* = 6*x* = 42), and a high repetitive character. The International Wheat Genome Sequencing Consortium (IWGSC) has carried out wheat genome assembly through a chromosome-based approach towards conquering the genome nature of wheat, but it only contained 10.2 Gbp of genomes with low contiguity [33,34]. Despite these efforts, the near-complete assembly of wheat has been achieved in a recent study [22].

This study demonstrates the possibility of assembling high complex genomes through a combination of sequencing Illumina short-reads and PacBio long-reads. The production of long-reads while using TGS is able to overcome the weakness of assembling short-reads by minimizing the generation of gaps or covering the repetitive sequence that appears on the plant genome. In another aspect, when only considering the accuracy, short-reads can be used for error-correction by aligning them to long-reads, which enable the increased accuracy of the genome assembly [35]. Therefore, a hybrid assembly through combinational sequencing is a useful approach, at least until now, for overcoming the limitations of the current two techniques. As a result, more accurate sequence data would be obtained. The MinION sequencing is expected to replace the PacBio sequencing in laboratory-level sequencing, even though the PacBio Single Molecule and Real-Time (SMRT) sequencing played a leading role, given the ease of performance and utilization of the MinION sequencing [29]. In the future, MinION sequencing [29] will play a significant role in noticeably improving the assembly of high complex genomes.

### 3.3. Improvements in the Accuracy of Long-Reads Sequencing and Assembly

We did not perform the Illumina sequencing in this study, since the sorghum genome sequence was already released. However, the *de novo* genome assembly is required for a reference-free species. Therefore, the hybrid assembly will be sufficient for overcoming the incompleteness that is caused by using a single platform. However, the hybrid assembly is more difficult than an assembly that uses a single platform. The genome assembly of plant species with a highly complex genome is possible if the accuracy of the raw long-reads is high, or it can be increased by using the MinION platform [29] alone. However, the accuracy of Nanopore sequencing (85% accuracy for R9 version) is not high when compared to that of the NGS generating short-reads [36]. Currently, even with the R9.5 flow cell using 1D^2^ chemistry for the MinION [29], the model accuracy of sequences that were obtained while using Nanopore sequencing is about 97% (http://nanoporetech.com) [29]. In contrast, the short-read from the Illumina platform has a maximum length of 150–300 bp, but most bases have more than 30 in quality score (99.9% accuracy) for single and paired-end reads (https://www.illumina.com). In addition, the improvement in sequencing accuracy can lead to the conclusion that the consensus accuracy will gain a high value from a small amount of raw read coverage [4]. For the *de novo* genome assembly, a raw read coverage of about 50–60× is needed to generate enough coverage of reads to cover repetitive regions in the genome assembly [37]. At this time, Nanopore sequencing for raw reads is not able to be more accurate than the accuracy of NGS, such as the Illumina sequencing, which produces highly accurate reads. Thus, an error-correction process is indispensable in increasing the accuracy in Nanopore sequencing. Because of this, if only MinION is used as a single platform, the significance of correction tools for raw reads is greater. Nanocorrect (https://github.com/jts/nanocorrect/), and PoreSeq [38] have been developed as a representative error-correction tool for the Nanopore sequence data. Recently, Canu [23], Falcon (https://github.com/PacificBiosciences/FALCON/), and Miniasm [24] assemblers are more commonly used for error correction, as well as the assembly. However, we should be aware that it is advantageous to obtain long-reads using HMW gDNA when sequencing using MinION [29] as a single platform, since adding reads to reduce the average length of reads is able to reduce the assembly’s quality [39].

### 3.4. DNA Fragmentation Effect on MinION Sequencing

The MinION flow cell (R9.4) that consists of 512 channels and four wells (four nanopores) is included in each channel. However, the read data are only generated from one of the four wells at a time [37]. From our results, the 2nd result that used fragmented gDNA produced more read data than the 1st and 3rd results that used intact HMW gDNA. However, the read length showed the opposite pattern. This may be due to the feature that the Nanopore sequencing could not be simultaneously performed in four wells in each channel. As a result, the following possibilities can be considered. First, in the process of tethering DNA molecules onto a membrane near a pore protein, HMW DNA molecules may cause the spatial hindrance to deteriorate the accessibility of the other DNA molecules to the nanopore. Second, the time that is required for the HMW gDNA molecule to pass through the nanopore is too long to allow for sequencing in the other wells of the same channel. This may result in a decrease in sequencing efficiency. For example, when using R9 chemistry (about 250 bp sequencing speed per second; https://nanoporetech.com), it takes about 1000 s for 250 Kbp of the DNA molecule to sequence. In this case, assuming that it shows 100% efficiency, the number of reads that were obtained through MinION [29] sequencing for 48 h is only 172.8 reads in each channel. For now, we need to choose whether to get a relatively large number of reads or to get reads that are as long as possible, depending on the experimental purpose. We expect to meet both through future technical advances.

### 3.5. Requirement of Effective Size Selection for Long-Reads Sequencing

It is important to remove the short-reads for high quality assembly. In this study, a large amount of short-reads (around 1 kb) was generated by MinION sequencing [29]. As aforementioned, the assembly quality can decrease if short-reads less than the average length are produced. Thus, we should consider the possibility of generating a lot of short-reads, even though HMW gDNA is used as an initial material. In general, a certain level of DNA supercoiling is maintained in vivo [40]. However, during DNA extraction, the DNA may be damaged, and the DNA supercoil level may decrease. After DNA extraction, DNA repair and adapter ligation steps are performed during the DNA library preparation for MinION sequencing [29]. At this time, the efficiency of library production may vary, depending on the structural complexity of the DNA. Highly ordered structures of genomic DNA may reduce the accessibility of enzymes that are involved in the DNA repair or adapter ligation, while short DNA fragments are expected to increase the efficiency of library production due to the relatively high accessibility of the enzymes. We also cannot rule out the possibility of DNA shearing by physical or chemical reactions during the DNA library preparation.

Another possibility is limiting the use of magnetic beads in the size selection and purification of the DNA library. Magnetic beads make it easy to remove small DNA that are less than 500 bp, but they are not effective in removing large size DNA. In addition, the yield of the DNA itself is greatly reduced when a relatively small amount of beads is used to obtain large-sized DNA fragments (e.g., DNA fragments of 1–10 kb in size). It is possible that the limitations of the protocols that are used in this study may not have effectively removed small size DNA. This can be overcome by conventional size selection methods, such as using gel electrophoresis and gel elution or automated DNA size selection (e.g., Pippin). However, until now, automated size selection is the most effective method, although it does not completely remove the short-reads. More accurate Nanopore sequencing and subsequent analysis will be possible if a more convenient and efficient size selection method is developed.

## 4. Materials and Methods

### 4.1. Plant Material and Genomic DNA Extraction

The sorghum reference accession BTx623 was obtained from the National Agrobiodiversity Center of the Rural Development Administration in Korea. Sorghum plants were grown on a Murashige and Skoog (MS) medium (Duchefa) in an artificial growth chamber (25 °C, 14 h light/10 h dark) for 7–10 days. Shoot parts were only used for genomic DNA (gDNA) extraction, and the procedure of the gDNA extraction was performed following the method that was previously described ([41,42]), with some modifications. Shoots of sorghum seedling were ground into a fine powder in liquid nitrogen while using a mortar and pestle. 100 mg of the sample powder was transferred into a 2 mL tube (eppendorf) containing 600 µL of a modified Carlson buffer [100 mM Tris-HCl, pH 8.0, 2% CTAB, 1.4 M NaCl, 1% PEG 8000, 20 mM EDTA, 2% PVP40, 0.1% ascorbic acid] pre-warmed to 60 °C and 20 µL of RNase A (20 mg/mL; invitrogen). The sample was immediately homogenized by inverting it gently 20 times and then incubating it in a water-bath at 60 °C for 30 min. with gentle inverting 20 times every 10 min. After incubation, the sample was cooled-down to room temperature, and 600 µL of chloroform was added. The sample was inverted carefully 60 times. Afterwards, the sample was centrifuged at 5000 g for 10 min. at 4 °C, and 400 µL of the supernatant was transferred to a new 2 mL tube. 400 µL of the binding buffer (20% PEG 8000, 3 M NaCl) and 50 µL of the AMPure XP beads solution were added to the sample and then incubated with rotation (6 rpm) at room temperature for 10 min. The sample was briefly centrifuged and kept on a magnetic rack (Thermo Fisher Scientific, Seoul, Korea) until the magnetic beads were completely separated. The supernatant was removed without disturbing the pellet, and then 1 mL of 70% ethanol was added to the pellet. The pellet was incubated in ethanol for 1 min., and the supernatant was removed. The ethanol washing step was repeated three times. After the ethanol was removed, the sample was air-dried for one min. The pellet was eluted while using a Buffer EB (Qiagen). At that moment, the amount of the Buffer EB was adjusted, so that the eluate concentration was 80 ng/µL or more. As a result, HMW gDNA with a size longer than at least 50 kb was obtained.

### 4.2. Preparation of Sequencing Library and MinION Sequencing

12 µg of HMW gDNA was fragmented while using a g-TUBE (Covaris) by centrifuging at 3170 g for 60 s. (Labogene 1730R; rotor GRF-M-m2.0-24). Of the three flow cells, the HMW gDNA of one flow cell was only fragmented, and the rest was used in its native state. The DNA library was prepared with the ONT Ligation Sequencing Kit 1D (SQK-LSK108), and the DNA preparation method was based on the “1D gDNA long reads without BluePippin protocol” that was provided by the Nanoporetech community (https://community.nanoporetech.com/protocols/1d-gdna-without-bluepippin/v/1/all_steps). A total of 2 µg of gDNA (80 ng/µL) was used to construct the DNA library in each flow cell. MinION sequencing was performed using a R9.4 SpotON flow cell (FLO-MIN106), and the default script “NC_48Hr_Sequencing_Run_FLO-MIN106_SQK-LSK108” from the MinKNOW program was used to run the sequencing. Finally, the read sequence files (fastq format) were obtained from the MinKNOW workflow.

Nanopore sequencing data have been deposited in the NCBI’s SRA database with the accession number of PRJNA544582.

### 4.3. MinION Raw Sequences Mapped against the Reference Genome

The generated raw fastq files from the MinKNOW workflow were mapped to the sorghum BTx623 reference genome (v.3.1.1) and downloaded from the plant genomics resource (https://phytozome.jgi.doe.gov/pz/portal.html) by using the BWA mem (version 0.7.15) [26] with default parameters.

The average depth was evaluated with a depth option, and the mapping rate was conducted with the flagstat option in SAMtools version 1.3.1 [28]. The Mosdepth program version 0.2.3 [27] was used to calculate the depth from the BAM file at each nucleotide position in a genome and to produce coverage graph (Figure 1).

### 4.4. De Novo Whole Genome Assembly

Some specially designed tools were adopted in order to efficiently handle these noisy long MinION reads. Two *de novo* assemblers were selected to compare their performances: Canu (version 1.6) [23] and Minimap and Miniasm (version 0.2-r168-dirty) [24]. Canu [23] is a new single-molecule sequence assembler that improves the Celera Assembler. Canu operates in three phases: Correction, trimming, and assembly. The correction step improves the accuracy of each read base. For the AT/GC rich eukaryotic genomes, the corMaxEvidenceErate = 0.15 parameter is suggested by the developer’s instructions. Therefore, this parameter was incorporated to run our sorghum data and other options were set as a default.

Minimap [24] with -Sw5 -L100 -m0 -t8 options and a *de novo* assembler, Miniasm [24] with default parameters were used to assemble MinION sequencing reads without an error correction stage. Minimap is an all-against-all read self-mapping tool, and Minasm is composed of simple concatenated pieces of the read sequence to generate the final unitig sequences. This tool allows for sequencing data to be assembled into a single contig in a relatively short time. However, the consensus sequence error rate is as high as the raw reads. Therefore, Racon (https://github.com/isovic/racon) [30], coupled with the Miniasm, could be used to generate similar or better quality final unitig sequences. Multiple rounds of Racon polishing have given a good final sequence accuracy and produced the best possible consensus sequence. To improve the sequence’s quality, we conducted five rounds of Racon [30] using Minimap and Miniasm [24] results.

### 4.5. Confirmation of de Novo Assembly against the Sorghum Reference Genome

We aligned the assembly results from the Miniasm against the sorghum BTx623 reference genome to test the structural correctness of the unitig genome. The nucmer option from the MUMmer software version 3.0 [31] was used to obtain an overview of the global alignment between the contigs and reference genome. In addition, the delta-filtering option with the -r and -q parameters were used to filter the alignment results. The mummerplot option from the MUMmer [31] was used to draw the dotplot.

The MUMmer sequence alignment package [31] was designed to detect the homology regions in the genome sequences. For a dotplot, the reference sequence is laid across the *x*-axis, while the query sequence is on the *y*-axis. Wherever the two sequences agree, a colored line or dot is plotted. The forward matches are displayed in red, while the reverse matches are displayed in blue.

## 5. Conclusions

Minion sequencing has developed rapidly in less than five years since the advent of TGS. Advances in its chemistry have elevated the speed and accuracy of sequencing, and the contiguity of genome assembly was improved by enabling long-reads. In particular, if a single laboratory can invest 1500 USD with a personal computer, a researcher can easily generate 10–15 Gbp of long-read sequencing information, constructing a fairly good draft reference genome of an organism with the genome size of 300 Mbp (considering 30–50× coverage commonly required for *de novo* assembly). These developments have enhanced the high utilization and value of genome assemblies for plant species with highly complex genomes. We showed the results of MinION sequencing [29] for the *S. bicolor* cv. BTx623, in which the accuracy and coverage of raw data against the reference genome changed during the process of error-correction, *de novo* assembly, and polishing. Our results not only illustrate the use of appropriate tools for genome assembly through MinION sequencing [29] in plant species, but they also provide information regarding the amount of raw data required for a more accurate genome assembly. This is expected to contribute to complete genome sequencing in a variety of plant species, including reference-free species.

## Figures and Tables

**Figure 1 plants-08-00270-f001:**
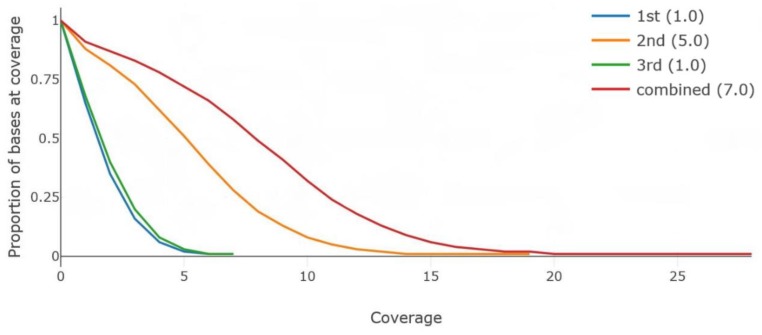
The coverage graph using Mosdepth. In this graph, the legend indicates the coverage graph for each result. The numbers in the parentheses indicate an average depth of coverage.

**Figure 2 plants-08-00270-f002:**
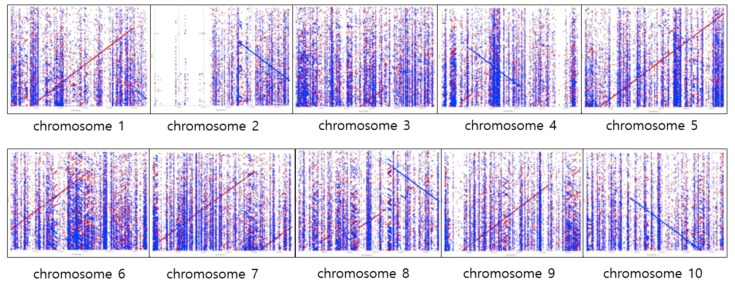
The five rounds polishing with Racon after Miniasm assembly versus each chromosome of the sorghum reference. The *x*-axis represents each chromosome of reference and the *y*-axis represents 2261 contigs. The forward matches are displayed in red, while the reverse matches are displayed in blue.

**Table 1 plants-08-00270-t001:** The statistics of the raw fastq file.

Result	1st	2nd	3rd
Total generated file size (Gb)	2.83	11.71	3.34
Total number of fastq files	35	170	37
Total read numbers	136,769	679,658	146,883
The shortest read length (bp)	167	74	38
The longest read length (bp)	190,250	110,486	217,000
The most abundant read length (bp) (no. of reads)	908 (61)	947 (111)	1028 (69)
Q-score	11.2	10.7	10.9

**Table 2 plants-08-00270-t002:** Results of average depth and mapping rate for raw reads against reference genome.

Result	1st	2nd	3rd	Combined ^a^
Average depth	2.01	5.64	2.10	8.56
Mapping rate (%)	97.93	96.87	97.14	97.08

^a^ Combined all three results.

**Table 3 plants-08-00270-t003:** Summary of read data for the results of Canu.

Result		1st	2nd	3rd	Combined
Total loaded reads	No. of reads	119,022	649,003	127,653	895,678
	Total length (bp)	1,495,987,647	6,216,312,936	1,767,114,081	9,479,414,664
	Coverage	2.04	8.51	2.42	12.63
Expected corrected reads	No. of reads	117,932	647,151	125,836	893,520
	Total length (bp)	1,333,102,902	6,187,551,664	1,359,065,630	8,029,184,425
	Mean read length (bp)	11,304	7,900	10,800	8,986
	N50 length (bp)	49,358	23,337	53,805	72,703
After correction/Before trimming	No. of reads	110,540	607,805	116,100	845,774
	Total length (bp)	1,235,198,760	5,658,532,542	1,549,842,529	8,673,782,926
	Coverage	1.68	7.75	2.12	11.56
After trimming ^a^	No. of reads	68,176	403,755	56,719	566,533
	Total bases (bp)	411,454,770	2,794,594,634	376,245,841	4,739,533,665
UniTigging/READs	No. of reads	68,176	410,746	56,719	577,103
	Total length (bp)	424,463,809	2,844,670,276	381,679,172	4,833,385,452
	Coverage	0.58	3.89	0.52	6.44
UniTigging/concensus	No. of sequences	159	5,740	127	6,124
	No. of repeats	28	692	26	712
	Length of repeats (bp)	573,105	10,509,344	472,695	14,815,759
	Total length (bp)	3,088,777	178,246,454	3,256,717	344,366,012
	Coverage	0.004	0.237	0.004	0.459
Unassembled	No. of sequences	38,897	168,888	32,340	216,120
	Total length (bp)	259,436,098	1,180,881,063	252,418,869	1,832,920,246

^a^ Trimmed reads output.

**Table 4 plants-08-00270-t004:** Summary of Miniasm assemblies with Minimap and Racon.

	Result	No. of Round	1st	2nd	3rd	Combined
Raw file	Total size (Gb)		2.83	11.71	3.34	17.9
Minimap	File size (byte)		607,227,298	5,089,824,937	546,909,744	13,226,110,131
Miniasm	File size (byte)		1,282,822	177,933,354	2,126,525	368,271,934
	Total length (bp)		1,286,782	176,978,175	2,139,682	370,303,449
Racon	Total length (bp)	1	1,289,492	177,650,167	2,145,749	373,675,134
		2	1,278,467	177,931,139	2,141,277	374,668,365
		3	1,262,947	177,915,228	2,127,089	374,934,532
		4	1,247,138	177,805,838	2,112,239	375,048,732
		5	1,232,808	177,683,528	2,097,341	375,105,174

**Table 5 plants-08-00270-t005:** Comparison between Canu and Miniasm using final assembly results.

	Canu	Miniasm
Number of Conigs	6124	2661
Assembled read length (bp)	344,366,012	375,105,174
N50 (bp)	98,000,000	199,000,000

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
