# Peer review of "Constructing a Reference Genome in a Single Lab: The Possibility to Use Oxford Nanopore Technology"

_plants, 2019, doi:10.3390/plants8080270_

Round 1

Reviewer 1 Report

Summary: Lee et al use minIon technology to generate a de novo assembly of the sorghum genome. MinIon is a promising technology, but there are few examples to date of its application to real projects such as this one. This is a strong proof of concept demonstrating how a forward-thinking lab can use this technology to make scientific achievements. This study will be most useful outside of sorghum, in species where reference genomes are not currently available.

However, there are a number of issues that should be addressed. I would like to see the authors make the code used public, as well as a more thorough investigation and demonstration that the assembled genome is of a decent quality. Additional major and minor suggestions are provided below.

Major:

Line 110: If genome coverage was 25X, but average read depth after alignment was only 8.6X, how can it be that the mapping rate was 97%? I am probably misunderstanding something, but I suspect other readers may also get confused about this without additional explanation.

Line 194-196: If it is worth stating that the authors could not formulate the relationship between coverage required and raw reads, the ‘various factors’ prohibiting this formulation should be stated also.

The comparison of the minION assembly to the existing sorghum assembly is not clear – Figure 2 gets at this, but there is little investigation into whether the overlap is good or bad. I think this needs to be elaborated upon.

The authors state that they have developed a pipeline to assemble de novo genomes from minION, but they do not make any code available. To enhance the utility of their publication, as well as to allow outside groups to evaluate the study for reproducibility, the code used should be made available.

Minor:

Line 38-39: The sentence refers to plants and animals, but the references refer mostly to humans. Why not include similar reviews in plants?

Line 51: What does it mean to monitor sequence information in real time?

Is it possible to give a sense of the cost for minIon? The manuscript discusses the ‘low’ cost, but as a person who has not used the technology I only know that the machine itself costs on the order of 1,000 USD -- I am completely unaware of the cost of consumables and library preparation, and I think this could be useful to mention at some level in this manuscript.

The manuscript should undergo English-language editing before publication.

Author Response

Summary: Lee et al use minIon technology to generate a de novo assembly of the sorghum genome. MinIon is a promising technology, but there are few examples to date of its application to real projects such as this one. This is a strong proof of concept demonstrating how a forward-thinking lab can use this technology to make scientific achievements. This study will be most useful outside of sorghum, in species where reference genomes are not currently available.

However, there are a number of issues that should be addressed. I would like to see the authors make the code used public, as well as a more thorough investigation and demonstration that the assembled genome is of a decent quality. Additional major and minor suggestions are provided below.

-       We are very thankful for the reviewer’s time and efforts for reviewing this manuscript. We tried to answer and correct the points that the reviewer suggested. We also added the comment that the raw data are deposited in the NCBI’s SRA with accession numbers for public use (Lines 374-375 and 438-439).

Major:

Line 110: If genome coverage was 25X, but average read depth after alignment was only 8.6X, how can it be that the mapping rate was 97%? I am probably misunderstanding something, but I suspect other readers may also get confused about this without additional explanation.

-       This part can make readers confused. To make things clear for the reviewer, we generated a total of 17.9 Gb data which covers about 25 times of sorghum genome (ca. 740 Mb). The average read depth after aligning them to sorghum reference genome was ca. 8.6x because a number of reads were focused on specific regions which could be repeats. The result indicates that ONT tends to intensively sequence repeated regions of genome, which is caused by the preparation of high molecular weight DNA. Still, the raw reads generated by ONT covered about 97% of sorghum genome, indicating that ONT can be affordable to generate draft genome data. The explanation of this point can be found in Lines 120-122 (However, the mapping rates were more than 97%, indicating that the sequencing data generated from the MinION platform could contain redundant coverage in the specific regions of the sorghum genome.)

Line 194-196: If it is worth stating that the authors could not formulate the relationship between coverage required and raw reads, the ‘various factors’ prohibiting this formulation should be stated also.

-       Some factors that can affect the genome assembly with long-read data are described in the “Discussion” (Lines 211-212)

The comparison of the minION assembly to the existing sorghum assembly is not clear – Figure 2 gets at this, but there is little investigation into whether the overlap is good or bad. I think this needs to be elaborated upon.

-       Although the raw data cover ca. 97% of sorghum genome, the assembled contigs from Canu and Miniasm covered only 45.9% and 50% of sorghum genome, respectively, clearly indicating that more than 25X raw data will be needed to complete the de novo assembly of sorghum genome. However, our main idea is to provide some useful information using ONT to construct a draft genome in a single lab for readers. We were supposed to have good coverage of contigs but it is limited by our funding situation at this point. But we are pretty sure that we would have better genome coverage if we can accumulate more data by additional ONT running, As the reviewer suggested, we added some sentences in the “Results” (Lines 188-190) and “Discussion” (Lines 206, 212-217).

The authors state that they have developed a pipeline to assemble de novo genomes from minION, but they do not make any code available. To enhance the utility of their publication, as well as to allow outside groups to evaluate the study for reproducibility, the code used should be made available.

-       To share codes for any bioinformatics project is indispensable. We are on the same page with the reviewer. We are ready to and willing to share any codes that we publish. However the pipeline in this manuscript means to use the series of bioinformatics tools that are already available in public. We tried to find an optimal combinations of those publicly available tools for better assemblies of long-read sequencing data. In fact, another reviewer also suggested the same point. Accordingly, we changed the word “pipeline” to another word which can best describe our intention to show in this manuscript (Lines 32, 33, 94, 96, 399, and 402). Also, If any group approaches us for this manuscript, we will share our experience to help our research community.

Minor:

Line 38-39: The sentence refers to plants and animals, but the references refer mostly to humans. Why not include similar reviews in plants?

-       We are thankful for the considerate suggestion. We replaced the reference #2 (Levy and Myers, 2016) to Apples et al. 2015 to cover all the organisms that we stated in the sentence.

Line 51: What does it mean to monitor sequence information in real time?

-       In the case of MinION sequencing, we can check out the progress of sequencing by time zone, using a program called MinKNOW. For example, information such as the number of reads generated in real-time and the length distribution of the reads during the sequencing process can be visually observed through the MinKNOW program. Accordingly, we corrected the sentence for better understanding (Line 57)

Is it possible to give a sense of the cost for minIon? The manuscript discusses the ‘low’ cost, but as a person who has not used the technology I only know that the machine itself costs on the order of 1,000 USD -- I am completely unaware of the cost of consumables and library preparation, and I think this could be useful to mention at some level in this manuscript.

-       The reviewer pointed out a very critical point of this manuscript. If we have a MinION device with a price of 1,500 USD and a computer, we can do easily experiment at a single laboratory level. For NGS and PacBio platforms, they require expensive equipment. We added a sentence in the “Conclusions” section, so that readers can budget and design their own experiment.

The manuscript should undergo English-language editing before publication.

-       We are served by an English editing company as the reviewer requested. The certification of English editing service is attached at the end of our manuscript.

Reviewer 2 Report

In this manuscript, the authors sequenced Sorghum genome by harnessing Nanopore technology, meanwhile, they try to assemble them using three softwares. Overall, the work is not prepared well to publish, include the scientific sounds and language. 

1) The authors mentioned "constructing a reference genome in a single lab", but they did not provide a new improved reference.

2) The authors mentioned that they provide a pipeline of long-read sequencing analysis, but no any pipeline was provide. This work just contain two stuffs: one is generating long reads using Nopopore sequencing machine, another is mapping the those long reads to public reference genome.

3) Even the authors can assemble the long-reads together based on the public reference genome as guide, the reads also contain lots of sequence error. That would be hard to adjust by the long-reads. So, they also need the short-reads to verify the sequence. Only long-reads is not sufficient to assemble a genome as reference, no matter the structure or base quality.

Author Response

# Reviewer 2

In this manuscript, the authors sequenced Sorghum genome by harnessing Nanopore technology, meanwhile, they try to assemble them using three softwares. Overall, the work is not prepared well to publish, include the scientific sounds and language. 

-       We are very thankful for the reviewer’s time and efforts for reviewing this manuscript. We tried to answer and correct the points that the reviewer suggested. Also, we are served by an English editing company as the reviewer requested. The certification of English editing service is attached at the end of our manuscript. We also added the comment that the raw data are deposited in the NCBI’s SRA with accession numbers for public use (Lines 374-375 and 438-439).

1) The authors mentioned "constructing a reference genome in a single lab", but they did not provide a new improved reference.

-       That is a good question to be pointed out. Basically we tried to test the possibility of using ONT for constructing a draft genome when it is needed. For GBS analysis example, the availability of reference genome (even if it is a draft) dramatically affects the genotyping results. If one can build up a draft genome with cheap price, the results will be much improved. In that point, we tried to provide some solutions to those who are working on any organism with no reference genome sequences. If the reviewer thinks that the title is not appropriate, we are willing to change it upon the reviewer’s request.

2) The authors mentioned that they provide a pipeline of long-read sequencing analysis, but no any pipeline was provide. This work just contain two stuffs: one is generating long reads using Nopopore sequencing machine, another is mapping those long reads to public reference genome.

-       We are grateful for the good point here. In fact, other reviewer pointed out the same thing. The pipeline in this manuscript means the series of handling long-read sequence data. We just tried to optimize the analysis of long-read data. Accordingly, we changed the word “pipeline” to another word which can represent our intention to show in the current study (Lines 32, 33, 94, 96, 399, and 402).

3) Even the authors can assemble the long-reads together based on the public reference genome as guide, the reads also contain lots of sequence error. That would be hard to adjust by the long-reads. So, they also need the short-reads to verify the sequence. Only long-reads is not sufficient to assemble a genome as reference, no matter the structure or base quality.

-       This is a critical point and we appreciate your opinion. The truth is that long-read data still has a number of sequencing errors with no corrections. The assemblers used in the current study have some error correction functions based on high coverage of data. For example, Canu has its own algorithm to detect sequencing errors and the combination of Miniasm and Racon provides similar functions. Consequently, if one organism is sequenced with ONT in high coverage, the long-reads data can generate an affordable draft genome without the aid of short-read sequencing. However, the data may not be suitable for genotyping population data due to those high error rates. In addition, we could not formulate the relationship between the minimum coverage required for the de novo assembly and the amount of raw reads because of various factors. Those factors were added in the “Discussion” (Lines 211-212)

Round 2

Reviewer 2 Report

The authors still not provide the new reference genome as supplemental information.

Author Response

Point 1: The authors still not provide the new reference genome as supplemental information.

Response 1: Reviewer made a good point which we didn’t catch up. We appreciate it. We tried to upload it in the public databases but it takes some time. To keep the deadline of revision, we saved the assembled contigs in our internal server and made it publicly available. One thing is that we asked our university to open a port to share the file outside our institution, which will take about 1-2 to business days. So you may not be able to see it at the point you are evaluating our manuscript. For your information, I also attached a screen capture to show you the contigs are available in our internal sever. Thank you very much for your kind suggestion. The comments are added in Lines 439-441.
